# Effect of provincial competitiveness index on enterprise attraction in the Central Highlands, Vietnam

Nguyen Phuong Le 👤*◉, Luu Van Duy 👤◉

Department of Agricultural Economics and Policies, Faculty of Economics & Rural Development, Vietnam National University of Agriculture, Hanoi, Vietnam

◉ These authors contributed equally to this work.
* nguyenphuongle@vnua.edu.vn, nguyenphuongle.vnua@gmail.com

## Abstract

This study aims to make a critical review of provincial competitiveness index (PCI) in the Central Highlands of Vietnam. In this regard, the research examines relationships between PCI and the possible attraction of enterprises operating in the region, thereafter, proposes some policy recommendations to improve business environment in the region. The secondary dataset of PCI and enterprise were incorporate with primary data collected from qualitative methods to explore the effect of PCI on enterprise attraction. The results show that the enterprises which have been operating in the Central Highlands accounted for 2.2% of the total with invested capital and accounted for 1.1% of whole country in 2018. PCI highly correlates with the number of enterprises, the employees and the capital of enterprises. Consequently, in order to attract enterprises, local governments in the region need to improve sub-indexes thereby improving general PCI scores.

## 1. Introduction

The Central Highlands or the Western Highlands, is located in the West and Southwest of Vietnam. The region contains five provinces: Lam Dong, Dak Lak, Dak Nong, Gia Lai and Kon Tum. These provinces are expected to show high potential for development in renewable energy, agriculture and tourism. Central Highlands cover an area of 5.46 million ha, in which 2 million ha are used for agricultural development. Especially, the region covers 74.25 percent of the total red basalt soil area of Vietnam, making it an ideal place for large-scale production specialized in coffee, pepper, tea, cashew, cassava, rubber. The Central Highlands contributed 9 percent of Vietnam's GDP in 2017 [1].

In recent years, central government and local authorities in the Central Highlands have issued and implemented varieties of policies and solutions to attract businesses in economic development in the region. Thanks to that, the Central Highlands economy has achieved many positive results, focusing on the development of a number of important domains. However, the region's growth is still modest as compared to other regions in the country. As of May 2018, the Central Highlands has about 15,000 enterprises, accounting for just 3% of the

**Data Availability Statement:** The data underlying this study are available from PCI (https://pcivietnam.vn/en/pci-data) following the protocols outlined in the Methods section.

**Funding:** The authors received no specific funding for this work.

**Competing interests:** The authors have declared that no competing interests exist.

total number of enterprises in the nation. The region's business density is also small (25 enterprises/10,000 people) as compared to the national average of 60 enterprises per 10,000 people. The flow of Foreign Direct Investment (FDI) into the region is still very limited: there are only 144 FDI registered companies, with total registered capital of USD 0.9 billion [2]. In addition, the firms mainly fall in small and extreme-small scales. The main reason is the fact that enterprises in the Central Highlands are facing with a number of difficulties in business environment such as market access, land access, low quality of human resources, lack of infrastructure and complicated legal procedures. It is witnessed that in 13 years (from 2006 to 2018), five provinces of the Central Highlands have been in the bottom rank of the Provincial Competitiveness Index (PCI) rankings of whole country. In 2018, only Gia Lai and Kon Tum have promoted, the remaining provinces have dropped, of which Dak Nong continued to be at the bottom (63/63) of country's ranking. Therefore, improving business environments to attract enterprises for economic development in the region becomes an urgent issue not only for particular province but also for whole region.

The paper aims: (i) to make an overview of the current PCI sub-indices in the Central Highlands and (ii) to specify how PCI points affecting enterprise development in the region, then (iii) to propose solutions to improve business environment in the Central Highlands.

## 2. Theoretical background

"Competitiveness" concept is as prominent as globalization [3]. Tomasz and Aldona in their study have reviewed sixteen concepts of "competitiveness", and eleven concepts and theories related to "competitiveness" [4]. It can be said that competitiveness is a multifaceted concept whose understanding comes from economics, management, history, politics and culture [5]. It has been described as a complex, multidimensional and relative concept, the relevance of which changes with time and context [6, 7]. Research with different backgrounds and particular purposes have attempted to study competitiveness, adding a different perspective in it. Competitiveness has become synonymous with economic strength, and it has been considered at different levels including nation, regional/local, industry, firm and product [8–10]. In this article, competitiveness is analyzed at national and local levels.

At national level, competitiveness has been paid much attention by a number of scholars. Most of them agreed that competitiveness of a nation is the ability of an economy to provide its population or residents with a rising living standards, high rates of growth in GPD per capita, and high rates of employment on a sustainable basis [11, 12]. Apart from that, national competitiveness has been also considered as the degree to which a country could produce goods and services meeting international market and occupying an increasingly world market share [13–15]. While all mentioned concepts of national competitiveness emphasized on specific economic indicators such as GDP per capita, rate of employment, goods and services share in the world market, some others focused on country's ability in providing business environment for its firms. In this regard, World Economic Forum [16] and Ha and Thu [17] defined competitiveness as the set of institutions, policies, and factors that determine a national efficiency and productivity. Whether an economy has a sustainable and prosperous development or not depends on the high or low national competitiveness, as well as the favorable or less favorable business environment.

To measure national competitiveness, several indices have been constructed with the Global Competitiveness Index (GCI) being the most widely accepted and discussed. The GCI has been firstly produced in 2004 by World Economic Forum, then it is published very year as the Global Competitiveness Report (GCR). The first index has been developed by Xavier Sala-i-Martin and Elsa V. Artadi and presented a holistic approach of national competitiveness [18].

After a transition period the GCR abandoned the previous indexes designed by Porter, Sachs and McArthur and from the 2006–2007 edition the GCI became the main tool to measure and compare national competitiveness [19].

The GCI of national competitiveness based on nine pillars that evolved into a twelve-pillar structure used until the 2017–2018 edition of the report. The last edition of the GCI before the major methodology change in 2018 was based on 114 indicators sorted into the 12 main pillars (Table 1). The twelve pillars were divided into three main groups (basic requirements, efficiency enhancers, and innovation and sophistication factors) to sort the participating countries that are on various levels of economic development into three distinct phases of national competitiveness–factor-driven, efficiency driven and innovation-driven [20].

Vietnam has implemented economic reform to transform the economy from central planning system into market-oriented one since 1986. Thanks to radical changes in policies, Vietnam's economy has dramatically grown and the country has moved from low to middle income group since 2008. Despite the significant economic improvement, Vietnam national competitiveness in the world remains low. In the 14-year period since the WEF began to assess national competitiveness by GCI (2006–2019), there has no substantial change in Vietnam competitiveness. The ranking of Vietnam's competitiveness has gone up and down, varying between 55[th] and 77[th] which traps the country in the middle group of surveyed economies.

Among ASEAN countries, Vietnam's competitiveness is also in the middle, ranking between 5[th] and 6[th] in the ten-member group: above Cambodia, Laos and Myanmar, below Singapore, Malaysia, Thailand and Indonesia, swapped Philippines's places for several times. One of reasons that causes Vietnam's low national competitiveness is slow improved business environment [23].

Business environment has been assessed by the Easy of Doing Business Index which has been annually published by World Bank as "Doing Business" since 2003. The index has been issued based on data collected from 10,000 questionnaires with 11 specific topics, including: 1) Starting a business; 2) Dealing with construction permits; 3) Getting electricity; 4) Registering property;

**Table 1.  The pillar structure of the GCI in the 2017–2018 and in the 2018 edition.**

| GCI in the 2017–2018 edition | GCI in the 2018 edition |
|---|---|
| *Basic requirements* | *Enabling Environment* |
| 1. Institutions | 1. Institutions |
| 2. Infrastructure | 2. Infrastructure |
| 3. Macroeconomic environment | 3. ICT Adoption |
| 4. Health and primary education | 4. Macroeconomic stability |
| *Efficiency enhancers* | *Human Capital* |
| 5. Higher education and training | 5. Health |
| 6. Goods market efficiency | 6. Skills |
| 7. Labor market efficiency | *Markets* |
| 8. Financial market development | 7. Product market |
| 9. Technological readiness | 8. Labor market |
| 10. Market size | 9. Financial system |
| *Innovation and sophistication factors* | 10. Market size |
| 11. Business sophistication | *Innovation ecosystem* |
| 12. Innovation | 11. Business dynamism |
|  | 12. Innovation capability |

Source: Adopted from World Economic Forum [21], World Economic Forum [22].

5) Getting credit; 6) Protecting investors; 7) Paying taxes; 8) Trading across borders; 9) Enforcing contracts; 10) Resolving insolvency; and 11) Employing workers (World Bank group, 2020).

Similar to national competitiveness, Vietnam's environment business has slowly and unstably improved. Globally, Vietnam moved from 115th in 2006 to 68th in 2018, 69th in 2019, then went down to 70th in 2020. The country has been in the middle of between 155 and 190 surveyed economies. In ASEAN region, Vietnam has been ahead of Cambodia, Laos and Myanmar, Indonesia and the Philippines. However, the country has been behind Singapore, Malaysia, Thailand and Brunei. Although in recent years, Vietnam government has required line ministries to remove irrelevant conditions which cause difficulties for business, the process has been sluggish. In addition, some ministries are reluctant to simplify business conditions [23].

The difficult business conditions have negatively impacted only national competitiveness, but also local competitiveness. The theories underpinned the concept of local competitiveness was developed in the late 1990s [11, 15]. The local competitiveness approach enables to develop a conceptual framework that covers all aspects of competitiveness process in a historically determined dynamic. A local competitiveness system is a physical, mental or cyberspace environment in which concepts operate as an overall supportive (or unsupportive) system [3]. According to Anh [24], local competitiveness is created by factors that make up operating businesses environment. The operating business environment is a combination of factors that affect the competitiveness of enterprise from opinion and attitudes to behavior, creativity and entrepreneurship. These factors can be divided into two main groups such as (i) quality of social infrastructure and political, legal, cultural, social, educational, and health institutions and (ii) economic institutions and policies such as fiscal policy, credit and economic structure.

This paper concentrates on a specific level of local competitiveness as "Provincial Competitiveness". In Vietnam, the Provincial Competitiveness Index (PCI) was designed by the Vietnam Chamber of Commerce and Industry (VCCI) with the support of the United States Agency for International Development (USAID) since 2005 in order to measure and evaluate quality of economic governance, convenience and friendliness of the business environment and administrative reform efforts of the provincial governments, thereby promoting the development of private sector.

PCI has been built up by 10 component indicators. The enterprises assess the local business environment through these 10 indicators. A province/city is considered as having a good business environment when it has: 1) Low entry costs for business start-ups; 2) Easy access to land and security of business premises; 3) Transparent business environment and equitable business information; 4) Minimal informal charges; 5) Limited time requirements for bureaucratic procedures and inspections; 6) Minimal crowding out of private activity from policy biases toward state, foreign, or connected firms; 7) Proactive and creative provincial leadership in solving problems for enterprises; 8) Developed and high-quality business support services; 9) Sound labor training policies; and 10) Fair and effective legal procedures for dispute resolution and maintaining law and order [25]. The detail contents of these ten PCI indicators are relatively similar to the Ease Doing Business Index. Although PCI reports have been recognized widely as reliable reference to evaluate efforts of the provincial governments in the business environment and administrative reform, the research on effect of PCI on enterprise attraction seems to be overlooked in existing literature. In this research, we aim to fill this gap by posing the relationship between PCI and and the development of enterprises in Central Highlands, Vietnam. The hypothesis of this research is that PCI scores will be improved when ten indicators become better, additionally, the higher province's PCI scores are, the more enterprises will be attracted and developed in the province (Fig 1).

Every four years, the PCI re-evaluates its methodology and recalibrates the index in order to ensure the PCI reflects recent changes in the business environment of Vietnam as perceived

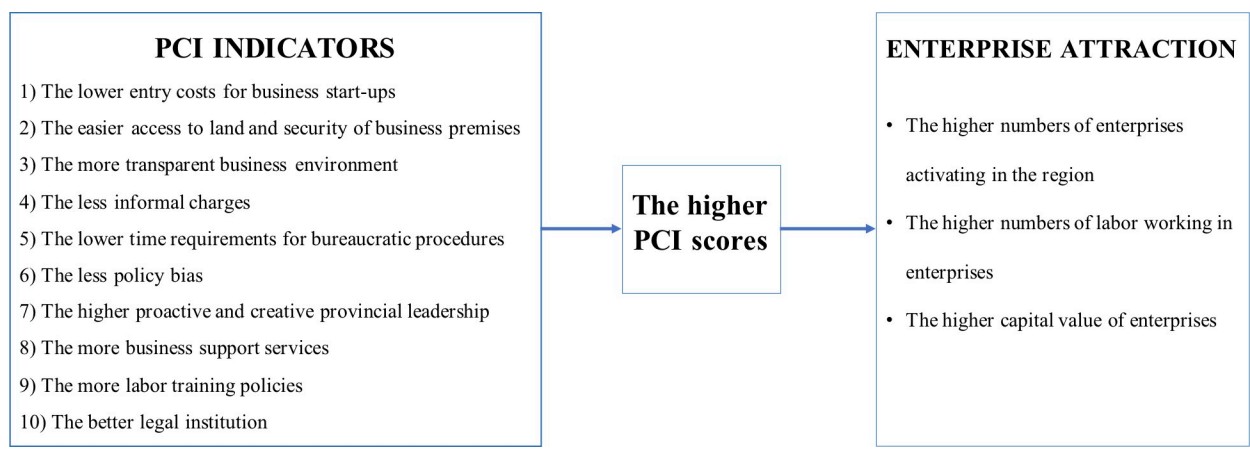

**Fig 1. The hypotheses of research.**

by businesses and provides a useful tool for policy makers. The current PCI has been adjusted in the most recent re-calibration in 2017, preceded by the ones in 2013 and 2009. PCI 2019 consists of ten sub-indices and a set of 128 indicators that were used in 2017 and 2018 and have been maintained in next year's report. The index is calculated in a three-step sequence such as: 1) collect business survey data and published data sources; 2) calculate ten sub-indices and standardize them on a 10-point scale; and 3) calibrate the composite PCI as the weighted mean of ten sub-indices with a maximum score of 100 points [2]. PCI of 63 provinces has been ranked based on total scores of ten sub-indices.

## 3. Research methods

### 3.1 Data sources

In this research, we collected secondary data from two sources. The first data source is from the PCI dataset which has been compiled annually since 2008 by VCCI under the support of USAID. The survey covers all 63 provinces of Vietnam. Annual survey has been carried out over 8,500 existing domestic private businesses that covers over 2,000 newly established enterprises and over 1,500 foreign-invested enterprises in all 63 provinces. Firms were selected by random sampling to mirror provincial populations. Stratification is used to make sure that firm age, size, legal type, and sector are accurately represented. The data contain 125,162 individual responses to questions asked each year in the annual PCI survey. Over 1,078 unique variables are covered. Based on the dataset of PCI at whole nation, we extracted PCI data of 5 provinces in the Central Highland region (i.e., Dak Lak, Dak Nong, Gia Lai, Kon Tum, and Lam Dong). The information of all ten PCI's sub-indices is available in the 2006–2019 period. There is only one the exception indicator "minimal crowding out of private activity from policy biases toward state, foreign, or connected firms" that have not been identified before 2013, therefore we analyzed the data in the 2013–2019 period.

The second data source is the status of enterprise development in the Central Highlands during 2006–2019 period which was collected mainly from the Statistical Yearbook of the General Statistics Office (GSO) and various publications of the Ministry of Planning and Investment (MPI). As it has not synthesis data of the status of enterprise in the Central Highland region yet, we tried to look for information from the GSO's reports, survey and any other notes. To make sure the validity of enterprise information, we contacted with staff of GSO, VCCI, and MPI with the hope that we can justify the precise of information.

Further, the primary data were collected through Focus Group Discussion (FGD) with the CEOs of enterprise in Dak Nong and Lam Dong provinces. Dak Nong were selected for primary data collection because it represents for the low PCI province, otherwise Lam Dong is reported as the high PCI province in the Central Highland. In doing so, 5–8 CEOs were selected for FGD in each province. The information from FGDs yielded understanding of the attitude and evaluation of firm owners with business environment and the quality of economic governance in the province. Moreover, 8 in-depth interviews were also undertaken with local authorities in Lam Dong and Dak Nong provinces. In these interviews, the information collected mainly related to information of status of enterprise development, the roles of local authorities in facilitating and improving business environment, and so on. The interview also aimed to collect the views and opinions of the provincial governments in term of administrative reform, policy supports, and the transparent business environment in the province. The interviews were conducted from June to September 2020. Direct quotes used in the analysis have been corrected grammatically for readability and all the names of the informants have been changed to ensure anonymity.

## 3.2 Data analysis

In this study, both qualitative and quantitative methods were applied to investigate the relationship between PCI and enterprise attraction in the Central Highland, Vietnam. The dataset of VCCI and GSO were entered into the Statistical Package for Social Sciences (SPSS). It was used for descriptive analyses to provide an overview of key trends of enterprise operating and CPI situation in the region. They were also used to cross-check, supplement themes and build arguments, along with the use of qualitative methods (e.g., in-depth interviews). Meanwhile, the data collected through qualitative methods were also used to prove the findings from statistical analyses.

Moreover, the correlation analysis was conducted to clarify the relationship between PCI points, its sub-indexes and the attraction of enterprises in each province and the whole region. In doing so, we estimate a sample correlation coefficient, more specifically the Pearson Product Moment correlation coefficient. The sample correlation coefficient, denoted r, ranges between -1 and +1 and quantifies the direction and strength of the linear association between the two variables. The correlation between two variables can be positive (i.e., higher levels of one variable are associated with higher levels of the other) or negative (i.e., higher levels of one variable are associated with lower levels of the other). The sign of the correlation coefficient indicates the direction of the association. The magnitude of the correlation coefficient indicates the strength of the association. A correlation close to zero suggests no linear association between two continuous variables.

## 4. Results and discussion

### 4.1 The enterprises operating in Central Highland region

The number of enterprises in the Central Highlands tends to increase by 6.2% per year in period 2011–2015. If comparing the period 2016–2017 with 2011–2015, the number of enterprises increased by 43.5%. In 2017, Dak Lak was a province with the largest number of enterprises in the Central Highlands, followed by Lam Dong, Gia Lai, Dak Nong and Kon Tum. In addition, the number of service enterprises in the region is relatively high, accounting for 67.7% of the total, followed by industry and construction with 28.2% and agriculture, forestry and aquaculture only 4.1%.

Non-state enterprises in the region accounted for 98.4% in 2017, an increase of 45.24% compared to the period 2011–2015. FDI enterprises have erratically changed. In 2017, FDI enterprises occupied only 0.5% of the total (much lower than the national average). State-owned enterprises in the Central Highlands tended to decrease rapidly (the rate of reduction was 24%/year), but they still account for 1% of the total (two times higher than the national average).

Total employees working in enterprises in the Central Highlands are 240,172 people in 2017, accounting for 1.65% of total labor force of the region (down 0.1% from the same period in 2016). In the 2016–2017 period, the number of employees working in enterprises in the region increase 0.2% compared to the period 2011–2015. The growth rate of employees is much slower than the growth rate of enterprises. This means that enterprises in the region cannot attract and create jobs for labors. Similar to general situation of the country, the number of employees working in industry and construction enterprises accounts for the largest proportion (41.3%), followed by service ones (37.6%) and agricultural ones (20.1%). On average, an agricultural enterprise employees 83.5 workers, an industry-construction enterprise employs 24 workers, whereas a service enterprise can only attract more than 9 workers [26].

Capital of enterprises in the Central Highlands accounts for about 1.1% of the whole country. The investment capital per agricultural enterprise in the region is 127.9 billion VND, two times higher than the national average (60.6 billion VND). The equity ratio of enterprises in the region is also much higher than the national average (42.5% and 28.4% respectively) [26].

Regarding to production and business results, in 2017, total net revenue of all operating enterprises reached to 334 trillion VND, an increase 10.1% compared to 2016. Agricultural enterprises in the Central Highlands generated 11.9 trillion of net revenue, accounting for 10.56% of total net revenue of agricultural enterprises nationwide, but only 3.56% of net revenue of enterprises in the region. Net revenue of agricultural enterprises in 2017 decreased compared to 2016 in provinces such as Dak Lak and Dak Nong (decreased by 18.2% and 7.3%, respectively) [26].

In summary, enterprises in the Central Highlands are not only few in quantity but also weak in capacity. Enterprises' labor and capital sizes are relatively small, but the land area is larger in comparison with other regions. Production results and efficiency of enterprises are low and tend to decrease in recent years. Many enterprises have suffered losses and workers' income has been decreased and much lower than the national average. As mentioned above, there are several reasons for this situation, but it can be said that unattractive business environment is the most important one. Particularly, almost 50% of interviewed CEOs have faced with difficulties in land area expansion and information accessibility. All of interviewed CEOs supposed that the SOEs always have been prioritized over non-state ones (*FGD, August 2020 in Dak Nong and Lam Dong*).

## 4.2 PCI sub-indices in Central Highland region

Fig 2 shows that there are 4 sub-indices (land access, transparency, policy bias, and business support services) where the average points of Central Highland provinces are equal to or higher than the national ones. Moreover, in the past ten years these sub-indices have significantly improved, especially the index of "land access" is rated higher by enterprises than other regions. Therefore, this article only focuses on analyzing the sub-indices that need to further reform in order to attract enterprises' investment such as: entry cost, time costs, informal charges, proactivity of provincial leadership, labor training and legal institutions.

**4.2.1 Entry cost and time cost.** Entry cost sub-index is used to measure how long a firm has to wait to register a business and to apply for a land use right certificate, and how long it takes to receive all required permits to conduct a business, including the number of licenses, registrations and required official approvals. The lower entry cost is, the better business environment. Fig 2 shows that the entry cost in the Central Highlands has erratically fluctuated. In the duration from 2013 to 2016, this sub-index has changed in a very positive direction. In 2013, the average score of this sub-index of the region was 0.02 points higher than that of whole country. Meanwhile in the three years 2014, 2015 and 2016, the regional score was only lower than that of the country 0.08 to 0.17 points. However, the gap between region's median of entry cost scores and the country's tended to increase in 2017 and 2018. In 2018, the

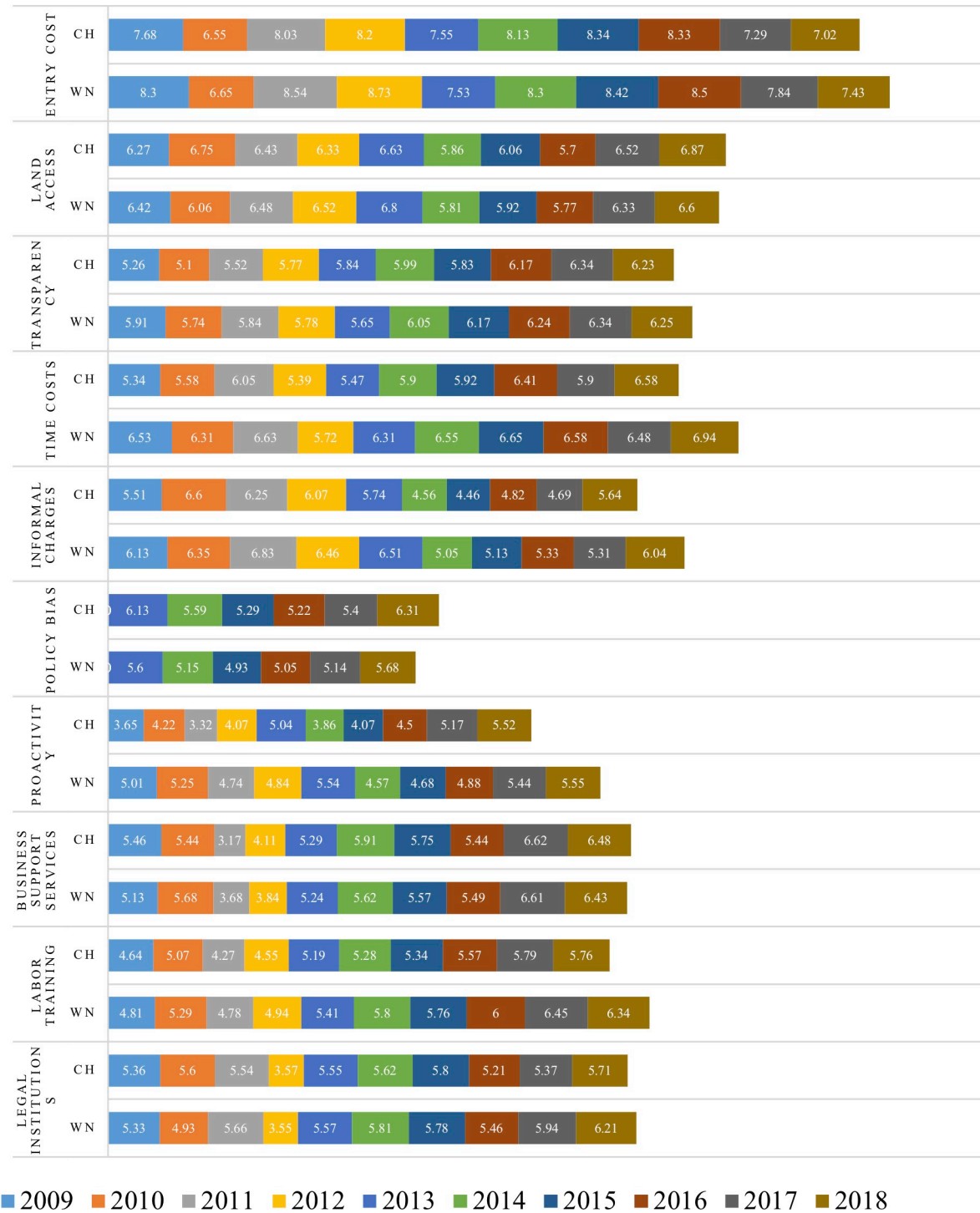

**Fig 2. Median of sub-indices points of the Central Highlands and whole country in 2009–2018 period.** *Note*: CH: Central Highland, WN: Whole nation. Unit: *points Source*: *Author's* compilation *from PCI database.*

enterprises who had to wait more than a month and more than 3 months to complete all the procedures for officially operating increased in comparison with the year 2016 (respectively 18.19% and 2.2% in 2016; 22.52% and 7.17% in 2018).

This finding is completely consistent with the sub-index of time cost. The score for time cost sub-index for the Central Highlands is lower than the national average score. The median gap of the Central Highlands and the whole country on the time cost tends to decrease but remains high, only in 2016, that gap dropped to the lowest of 0.17 points. While some indicators of time cost sub-index have been significantly improved (more efficient state officials, reduced number of hours worked with tax inspectors, civil servants friendlier so forth), the important indicators that reflect time costs have not changed much, especially the proportion of businesses who though they still have to go back and forth many times to complete administrative procedures are still high in recent years (from 30% to 40%).

The "time cost" sub-index of Central Highland provinces indicate local limitations in administrative procedure reform. Although some provinces have made progress, they still need to further reform and maintain achievements to facilitate business environment. Along with the review of all procedures, provinces need to pay more attention to the coordination among line departments in order to create favorable conditions for businesses.

**4.2.2 Informal charges.** This sub-index shows how much firms have to pay informal charges, how much of an obstacle those extra fees pose for their business operations, whether payment of those extra fees results in expected "services," and whether provincial officials use compliance with local regulations to extract rents. Fig 2 shows that the gap between Central Highland provinces and the rest of country from 2009 to 2018 have decreased, but it is still quite large (it is 0.4 points in 2018). This proves that the initial reforms in the provinces in the region have been effective, but this process has not been extensive enough, so the interference with enterprises has not been significantly reduced. It is reconfirmed by data in Table 2.

For a recent decade, among five indicators for assessing informal charges, the enterprises who had to spend more than 10% of their revenue on expenses have sharply dropped in 4 provinces except for Kon Tum (in 2018, the proportion of enterprises having to spend more than 10% of their revenue on informal charges in Kon Tum is two times higher than the national average). Other indicators have not changed significantly. Especially, the enterprises who supposed that their work will be resolved after paying informal costs tended to increase (except for Dak Nong province).

According to some experts, "informal charges" is a sub-index that can recover quickly in the short term. If administrative procedures are radically reformed, enterprises will no longer spend money for getting necessary documents. In addition, the stricter regulations are made by government, the lower corruption and informal charges are paid by enterprises [27].

**4.2.3 Proactivity of provincial leadership.** The sub-index "proactivity of provincial leadership" is used to measure the creativity of provincial leaders in the process of manipulating and implementing central government's policies by introducing initiatives to develop private sector, to assess ability of the central government's support and to apply policies in the direction of business facilitation.

Information in Table 2 shows that the median difference of "proactivity of provincial leadership" sub-index between the Central Highlands and the whole country significantly decreased (from 1.4 points in 2011 to 0.03 points in 2018). The provincial leaders' proactivity has been better evaluated by enterprises. A number of enterprises agree that provincial officials are creative in solving emerging problems in business. The proportion of enterprises satisfied with provincial governments' resolution sharply increased in all provinces in the region. However, Central Highland provinces still have to continue building staff capacity, improving human resource quality to meet needs of businesses development.

**4.2.4 Labor training.** The "labor training" sub-index measures provinces' efforts to promote vocational training and skills in order to support local industries and to help labors find jobs. The median scores of "labor training" sub-index of Central Highland provinces

**Table 2. Assessment of informal costs by enterprises in the Central Highlands.** *Unit: Percent of agreed enterprises.*

| Provinces | Enterprises in the same industry have to pay for informal charges | Percentage of firms paying over 10 percent of their revenue for informal charges | Rent-seeking phenomenon is popular in handling administrative procedures for businesses | Percentage of firms saying that informal charges usually or always deliver expected results | Informal charges are at acceptable levels |
|---|---|---|---|---|---|
| Dak Lak | | | | | |
| 2009 | 58.12 | 11.84 | 50.44 | 51.52 | 48.98 |
| 2018 | 58.42 | 2.35 | 65.49 | 69.14 | 48.39 |
| Dak Nong | | | | | |
| 2009 | 68.00 | 15.25 | 65.33 | 69.01 | 63.77 |
| 2018 | 54.55 | 6.93 | 60.19 | 62.22 | 55.17 |
| Gia Lai | | | | | |
| 2009 | 61.34 | 18.75 | 60.78 | 51.49 | 59.18 |
| 2018 | 53.85 | 6.12 | 66.67 | 62.20 | 60.00 |
| Kon Tum | | | | | |
| 2009 | 59.38 | 15.66 | 50.83 | 54.08 | 45.83 |
| 2018 | 61.86 | 15.91 | 67.02 | 69.23 | 55.17 |
| Lam Dong | | | | | |
| 2009 | 73.60 | 8.51 | 55.46 | 50.89 | 55.34 |
| 2018 | 50.00 | 5.77 | 50.85 | 58.70 | 44.00 |
| Whole country 2009 | | | | | |
| Mean | 59.41 | 8.75 | 50.35 | 51.52 | 53.47 |
| Max | 77.47 | 20.78 | 71.64 | 69.01 | 74.81 |
| Min | 35.38 | 2.61 | 23.93 | 35.42 | 22.88 |
| Whole country 2018 | | | | | |
| Mean | 54.81 | 7.08 | 58.16 | 61.64 | 48.39 |
| Max | 73.53 | 25.56 | 68.79 | 82.14 | 68.42 |
| Min | 38.32 | 1.68 | 37.32 | 46.27 | 14.29 |

Author's compilation from PCI database.

have been lower than the whole country's and tended to increase in recent ten years. This indicated that these provinces have not done well in training and job seeking for local labors (Table 3).

Similar to general trend of the country, most of indicators relating labor training have been improved in Central Highland provinces, especially the proportion of enterprises using labor exchange services of provincial agencies and the percentage of business agreeing labor meets firm needs. The costs spent on labor training in total enterprises' business expenses in region have increased and are higher than national average (except for Dak Lak and Kon Tum provinces). However, percentage of trained labors working in enterprises is very low (almost 50 percent. Moreover, the rate of enterprises who appreciated vocational training quality of the region is still low (from 25% to approximately 40%). Therefore, in order to meet the needs of enterprises, labor training and employment services in the Central Highlands must be renewed.

**4.2.5 Legal institutions.** The "legal institutions" sub-index is a measure of the private sector's belief in provincial legal institutions whether firms regard provincial legal institutions as an effective vehicle for dispute resolution or as an avenue for lodging appeals against corrupt

**Table 3. Indicators in "labor training" sub-index in the Central Highlands.** Unit: %.

| Time and Location | Percentage of business evaluating Services provided by provincial agencies are good | Percentage of enterprise used labor exchange services | Percentage of total business costs spent on labor training | Percentage of business agreeing labor meets firm needs | Percentage of workers having completed training at vocational schools | Percentage of workers having completed training at vocational schools working in enterprise |
|---|---|---|---|---|---|---|
| Dak Lak | | | | | | |
| 2014 | 25.00 | 32.35 | 4.62 | 92.86 | 7.88 | 38.13 |
| 2018 | 36.45 | 50.94 | 4.83 | 97.30 | 7.10 | 52.02 |
| Dak Nong | | | | | | |
| 2014 | 20.65 | 30.00 | 5.99 | 86.67 | 4.15 | 39.91 |
| 2018 | 25.96 | 63.98 | 8.29 | 85.00 | 5.85 | 45.47 |
| Gia Lai | | | | | | |
| 2014 | 23.60 | 36.25 | 3.43 | 95.51 | 4.42 | 34.28 |
| 2018 | 37.50 | 50.00 | 6.75 | 90.10 | 3.73 | 44.89 |
| Kon Tum | | | | | | |
| 2014 | 22.94 | 21.35 | 5.39 | 94.44 | 5.57 | 44.26 |
| 2018 | 29.47 | 90.91 | 4.87 | 84.27 | 6.93 | 44.64 |
| Lam Dong | | | | | | |
| 2014 | 31.34 | 39.52 | 5.11 | 90.63 | 7.10 | 44.62 |
| 2018 | 46.69 | 61.76 | 6.04 | 92.23 | 8.57 | 46.73 |
| Whole country 2014 | | | | | | |
| Mean | 33.08 | 27.94 | 5.56 | 93.59 | 7.61 | 42.29 |
| Max | 57.47 | 47.78 | 8.09 | 100.00 | 12.86 | 55.06 |
| Min | 16.30 | 13.11 | 3.38 | 77.46 | 2.14 | 23.19 |
| Whole country 2018 | | | | | | |
| Mean | 38.10 | 65.52 | 5.44 | 90.08 | 8.40 | 47.78 |
| Max | 62.24 | 90.91 | 8.42 | 97.92 | 13.53 | 56.54 |
| Min | 19.82 | 41.38 | 2.38 | 79.27 | 2.68 | 32.83 |

Source: Author's compilation from PCI database.

official behavior. The median scores of "legal institutions" sub-index in the Central Highlands are approximately equal to the country's ones. In the period 2009–2018, there were 3 years that the regional median scores were higher than the national average (Table 4). However, in 2017 and 2018, the score for this sub-index in the Central Highlands decreased. The data for 2018 show that indicators of the "legal institutions" sub-index of the country have been significantly improved, while in the Central Highlands, only Gia Lai province has six indicators which have been improved, all indicators have been worse in remaining 4 provinces. Especially, business-men did not believe that the legal system could help them denounce corruption.

Kon Tum is one of five provinces where the "legal institutions" sub-index is lower than the region's and country's medians. This partly reflects low confidence of enterprises in the provincial court and judiciary system. Therefore, in the coming years, Kon Tum authorities must make more efforts to build trust with businesses, so that they can see these legal institutions as a tool to resolve disputes or problems that they have to face with.

## 4.3 Relationship between PCI and the development of enterprises in the Central Highlands

**4.3.1 PCI and the rising of enterprises in the Central Highlands (Box 1).** Alongside sub-indexes and PCI points, there are many other factors which affect enterprise attraction to

**Table 4. Indicators in "legal institutions" sub-index in the Central Highlands.** Unit: %.

| Time and Location | Percentage of enterprises believing in legal system | Legal system could help enterprises denounce corruption | Willingness to use court in case a dispute arises | Judgement by the court is fair | Court judgements are enforced quickly | Legal aid agencies support business to use laws to sue when disputes arise |
|---|---|---|---|---|---|---|
| Dak Lak | | | | | | |
| 2015 | 78.91 | 31.97 | 40.32 | 72.73 | 65.79 | 70.80 |
| 2018 | 82.24 | 29.90 | 55.08 | 85.44 | 77.14 | 72.28 |
| Dak Nong | | | | | | |
| 2015 | 75.86 | 30.68 | 34.88 | 76.32 | 56.41 | 76.32 |
| 2018 | 78.26 | 29.47 | 45.16 | 76.47 | 65.88 | 67.86 |
| Gia Lai | | | | | | |
| 2015 | 80.19 | 21.78 | 42.72 | 68.75 | 65.26 | 69.15 |
| 2018 | 80.77 | 37.74 | 49.52 | 76.14 | 68.60 | 71.26 |
| Kon Tum | | | | | | |
| 2015 | 81.05 | 27.47 | 33.33 | 78.16 | 65.88 | 72.94 |
| 2018 | 78.82 | 17.71 | 34.78 | 78.26 | 67.00 | 72.83 |
| Lam Dong | | | | | | |
| 2015 | 82.28 | 30.38 | 36.36 | 79.14 | 66.42 | 65.22 |
| 2018 | 89.66 | 32.73 | 39.32 | 87.76 | 67.00 | 70.10 |
| Whole country 2015 | | | | | | |
| Mean | 81.20 | 31.39 | 37.50 | 81.98 | 65.26 | 72.15 |
| Max | 88.89 | 47.67 | 50.00 | 90.67 | 76.32 | 86.30 |
| Min | 70.36 | 18.29 | 23.42 | 68.75 | 50.00 | 55.95 |
| Whole country 2018 | | | | | | |
| Mean | 84.78 | 32.14 | 45.16 | 83.65 | 69.66 | 72.82 |
| Max | 93.33 | 51.65 | 60.00 | 93.67 | 83.51 | 87.95 |
| Min | 76.15 | 17.71 | 34.78 | 76.14 | 55.88 | 59.11 |

Source: Author's compilation from PCI database.

invest in the region such as: weak and inadequate physical infrastructures (electricity, water, information technology, terminals, regional and inter-regional traffics); social infrastructure has not met the needs of investors, especially the banking system, education, health care and other support services; the promotion of investors is not effective and the support for investors after licensing is not good [28].

### Box 1. Influence of PCI on enterprise attraction in Lam Dong province

In recent years, Lam Dong province has made great efforts in encouraging, supporting and creating favorable conditions for enterprise attraction. Thanks to the improvement of the business environment, the number of newly established enterprises has increased by an average of over 15% annually. According to interviewed CEOs in Lam Dong, they have been supported by several policies such as credit accessibilities, land renting, science and technology transferring, start-up support, labor training, and participating policy dialogue held by local authorities.

*Source*: *FGD, February 2019, Lam Dong province*

The results of correlation analysis between the business environment (e.g., sub-indexes and PCI points) and the number of enterprises in the Central Highlands from 2006 to 2018 has been indicated in Table 5. It is clearly seen that the index "transparency" is positively correlated with the number of firms with statistically significant at 1%. In other words, provinces rated better by firms in "transparency" tend to attract more the number of enterprises. This index reflects that the enterprises in the market need to access accurate and timely information sources to participate in transactions. Information from the State also plays an important role in reducing risks for businesses. Moreover, "time costs", and "labor training" are also positively correlated with the number of firms at 1% of statistically significant, meanwhile the "legal institutions" is positively correlated with statistically significant at 5%.

On the other hand, "informal charges" were negatively correlated with the number of firms at the 1% of statistical significance. These findings are in the same line with results in the PCI 2018 Report which pointed out that one point of improvement in this sub-index could lead to a 15% increase in new registrations over the next 10 years [25].

Remarkably, "land access" was not significant correlation with the number of firms. This finding is inverse with the results indicated by PCI 2018 Report in which showed that one increasing point in "land access" would increase the number of newly registered firms by 12% [25]. This manifestation could be explained by land scarcity are not much problem effecting to the new entry of enterprises in the Central Highlands.

**4.3.2 PCI and the number of labor of enterprises in the Central Highlands.**   Regarding the relationship between PCI and labor in the enterprises, the research found that "transparency" and "time cost" have positive correlations with the labor number of the enterprises in the region and those relations reflects high coefficient and statistical significance at the p-value < 0.05 (Table 6). The results indicate that the more effort from provincial governments to increase ease of access to information, the more attraction of employees working in the enterprises with an exceptional observation of Gia Lai province. Similarly, "labor training" has high correlation coefficient with the number labor of firms at the p-value < 0.01. It suggests that "labor policy training" in each province plays very important role in attracting the wage labor participating in the private enterprises. Thus, the employment policies should attempt to enhance the knowledge and skills for labor through government support programs such as training course for labor, support for finding a proper job, and so on.

**Table 5.  Correlation between PCI and the number of enterprises in the Central Highlands in the 2006–2018 period.**

| Indicator | Dak Lak | Dak Nong | Gia Lai | Kon Tum | Lam Dong | Whole region |
|---|---|---|---|---|---|---|
| Entry cost | 0.068 | 0.458 | -0.388 | -0.297 | 0.168 | 0.065 |
| Land access | 0.47 | 0.319 | 0.423 | 0.137 | -0.318 | 0.339 |
| Transparency | 0.386 | 0.846** | 0.139 | 0.705** | 0.830** | 0.941** |
| Time costs | 0.581* | 0.791** | 0.715** | 0.827** | 0.790** | 0.838** |
| Informal charges | -0.382 | -0.803** | -0.584* | -0.493 | -0.601* | -.686** |
| Policy bias | 0.372 | -0.239 | -0.025 | 0.153 | -0.01 | 0.111 |
| Proactivity of provincial leadership | 0.234 | 0.284 | -0.192 | 0.555* | 0.534 | 0.51 |
| Business support services | 0.241 | 0.481 | 0.327 | 0.354 | 0.154 | 0.424 |
| Labor policy | 0.729** | 0.651* | 0.489 | 0.865** | 0.866** | 0.836** |
| Legal institutions | 0.769** | -0.231 | 0.477 | 0.404 | 0.674* | 0.564* |
| PCI points | 0.874** | 0.810** | 0.714** | 0.874** | 0.882** | 0.929** |

Note: *, ** Indicate statistical significance at the p-value < 0.05 and 0.01, respectively.

Source: Author's calculation from secondary data, 2020.

**Table 6. Correlation between PCI and the number of labor of enterprises in the Central Highlands in the 2006–2018 period.**

| Indicator | Dak Lak | Dak Nong | Gia Lai | Kon Tum | Lam Dong | Whole region |
|---|---|---|---|---|---|---|
| Entry costs | 0.064 | 0.654* | 0.048 | -0.269 | 0.078 | 0.215 |
| Land access | 0.239 | 0.393 | 0.008 | 0.315 | -0.302 | 0.481 |
| Transparency | 0.215 | 0.908** | -0.712** | 0.562* | 0.863** | 0.779** |
| Time costs | 0.139 | 0.756** | 0.316 | 0.854** | 0.817** | 0.791** |
| Informal charges | 0.429 | -0.738** | -0.451 | -0.183 | -0.547 | -0.51 |
| Policy bias | 0.47 | -0.72 | 0.264 | -0.351 | 0.087 | 0.208 |
| Proactivity of provincial leadership | 0.031 | 0.266 | -0.384 | 0.311 | 0.505 | 0.156 |
| Business support services | -0.092 | 0.458 | -0.358 | 0.272 | 0.052 | 0.209 |
| Labor policy | 0.039 | 0.552 | 0.169 | 0.738** | 0.840** | 0.662* |
| Legal institutions | -0.199 | -0.331 | 0.426 | 0.434 | 0.741** | 0.589* |
| PCI points | 0.233 | 0.839** | -0.125 | 0.836** | 0.884** | 0.826** |

Note: *, ** Indicate statistical significance at the p-value < 0.05 and 0.01, respectively.

Source: Author's calculation form secondary data, 2021.

The results from Table 7 also demonstrate that PCI points positively correlated with the labor quantity in the firms in all provinces as well as whole region. The same effect is found in the PCI Report 2018, specifically, an increase of 1 point PCI would increase 7% of number of employees involving in the labor market of the province [25]. Hence, it can be concluded that the PCI index and its sub-indexes reflects the high relationship with the attraction of employees in each province and the whole region as well in the Central Highlands.

**4.3.3 PCI and the capital of enterprises in the Central Highlands.** Table 7 shows the correlation analysis results between the business environment (e.g., sub-indexes and PCI points) and the capital of enterprises in the Central Highlands during 2006–2018. Alike previous results, "transparency", "time costs" and "labor polices" have positive correlations with the size of the firm's asset at statistically significant of 1%. Otherwise, "informal charges" showed the negative correlation at the p-value < 0.05. These findings are completely relevant with PCI 2018 Report [25].

In general, PCI index and its sub-indices highly relate to the indicators reflecting the development of enterprises in the Central Highlands. The provinces with better PCI points tend to

**Table 7. Correlation between PCI and the capital of enterprises in the Central Highlands in the 2006–2018 period.**

| Indicator | Dak Lak | Dak Nong | Gia Lai | Kon Tum | Lam Dong | Whole region |
|---|---|---|---|---|---|---|
| Entry cost | 0.18 | 0.617* | -0.478 | -0.422 | 0.063 | 0.015 |
| Land access | 0.543 | 0.531 | 0.507 | -0.024 | -0.254 | 0.331 |
| Transparency | 0.326 | 0.963** | 0.469 | 0.617* | 0.764** | 0.935** |
| Time costs | 0.631* | 0.724** | 0.565* | 0.658* | 0.824** | 0.792** |
| Informal charges | -0.435 | -0.731** | -0.488 | -0.676* | -0.54 | -0.679* |
| Policy bias | 0.455 | -0.528 | -0.257 | 0.197 | 0.205 | 0.217 |
| Proactivity of provincial leadership | 0.135 | 0.326 | -0.153 | 0.731** | 0.570* | 0.528 |
| Business support services | 0.19 | 0.278 | 0.412 | 0.422 | 0.165 | 0.399 |
| Labor policy | 0.800** | 0.531 | 0.613* | 0.829** | 0.859** | 0.845** |
| Legal institutions | 0.765** | -0.298 | 0.234 | 0.218 | 0.618* | 0.483 |
| PCI points | 0.929** | 0.878** | 0.851** | 0.739** | 0.872** | 0.922** |

Note: *, ** Indicate statistical significance at the p-value < 0.05 and 0.01, respectively.

Source: Author's calculation from secondary data, 2021.

attract more enterprises, employees and the investment capital. Among sub-indices of PCI, "transparency", "time costs", "labor training" and "informal charges" are the most ones influencing on the attraction of enterprises investing into the provinces.

## 5 Conclusion and recommendation

Central Highland provinces have plenty of advantages in attracting businesses investment such as large land area, abundant natural resources and young labor force. However, the number of enterprises and total capital which invested in the region are still very low compared to other regions. The reason is the fact that local competitiveness is very low in comparison with other regions in the country. It is witnessed that PCI scores of the Central Highlands within 13 years (2006–2018) were in the second half of country's ranking. They were erratically fluctuated and slowly improved. Besides some highlighted sub-indices, particularly "access to land", there were others that showed limitations of provincial competitiveness in the Central Highlands such as entry cost, time cost, informal charges, labor training and legal institutions. The quantitative analysis showed that improvement of PCI points positively impacted on attraction of enterprises, employees and investment capital in the region. In order to improve competitiveness and further to promote enterprise attraction, it is necessary to implement solutions in coming years in the region including: (i) Continue to further improve and more strongly reform public administrative procedures, with a focus on simplification of administrative procedures, increasing online public services, through which minimizes the waiting time and informal charges of businesses; (ii) Reform vocational training system (vocational school system, vocational training programs and methods) towards better meeting the needs of the labor market, especially the needs of enterprises in the region; (iii) Renovate and enhance role of judicial system: on the one hand, promoting the role of commercial arbitration centers in resolving corporate disputes to avoid overloading of courts and shorten time, on the other hand to increase the court's support for commercial arbitration centers to ensure state power in dispute solving.

## Author Contributions

**Conceptualization:** Nguyen Phuong Le.

**Data curation:** Nguyen Phuong Le, Luu Van Duy.

**Formal analysis:** Nguyen Phuong Le.

**Investigation:** Nguyen Phuong Le.

**Methodology:** Nguyen Phuong Le, Luu Van Duy.

**Writing – original draft:** Nguyen Phuong Le.

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
