## [Decision Letter · Decision Letter 0]

30 May 2021

PONE-D-21-15678

Effect of Provincial Competitiveness Index on Enterprise Attraction in the Central Highlands, Vietnam

PLOS ONE

Dear Dr. Nguyen Phuong Le,

Thank you for submitting your manuscript to PLOS ONE. After careful consideration, we feel that it has merit but does not fully meet PLOS ONE’s publication criteria as it currently stands. Therefore, we invite you to submit a revised version of the manuscript that addresses the points raised during the review process.

We look forward to receiving your revised manuscript.

Kind regards,

László Vasa, PhD

Academic Editor

PLOS ONE

Journal Requirements:

2. We note you have included a table to which you do not refer in the text of your manuscript. Please ensure that you refer to Tables 1, 4 and 5 in your text; if accepted, production will need this reference to link the reader to the Table.

Reviewers' comments:

Reviewer's Responses to Questions

**Comments to the Author**

1. Is the manuscript technically sound, and do the data support the conclusions?

Reviewer #1: Yes

Reviewer #2: Partly

2. Has the statistical analysis been performed appropriately and rigorously? 

Reviewer #1: Yes

Reviewer #2: No

3. Have the authors made all data underlying the findings in their manuscript fully available?

Reviewer #1: Yes

Reviewer #2: Yes

4. Is the manuscript presented in an intelligible fashion and written in standard English?

Reviewer #1: Yes

Reviewer #2: Yes

5. Review Comments to the Author

Reviewer #1: The topic of is very interesting and also relevant. The elaboration and the applied methodology seem accurate. The tables however possibly should not lean to the following pages – it is just a matter of technical edition.

The bibliographic review is a bit narrow in depth and in broadness: In one hand, it should be broadened towards the experiences and lessons of other sub Asian countries, and in the other hand, the authors should utilize more international sources. In fact in the paper in its present form there is only one non-Vietnamese source (Porter, 2008), which is far insufficient in a high-ranking international journal. The international bibliography is really rich in this respect (e.g. https://www.ersj.eu/journal/2104/download or https://www.aljazeera.com/economy/2021/5/6/climate-change-brings-gains-for-some-vietnam-farmers-at-a-cost )

In the conclusion part the suggestions/proposals applicable for the further improvement are merely based on the research results and look impressive, however they could be elaborated more in details as they look a bit too general in their present form.

After the mentioned improvements were done, the paper can be recommended to be accepted and published.

Reviewer #2: The title of the paper is corresponded and the abstract is relevant to the content of paper.

The paper divided into several chapters. The conceptual framework and terminology that is required to present the analysis are consistent and in line with the conventions of the given scientific field.

A research hypothesis is a specific, clear, and testable proposition or predictive statement about the possible outcome of a scientific research study. A hypothesis is a tentative statement about the relationship between two or more variables. Specifying the research hypotheses is one of the most important steps in a scientific research study. This publication does not contain any hypotheses.

This research work is a secondary data analysis, as was written in the “Methodology" section. The lone primary research result of the paper is the correlation analysis. The authors should use more intensive primary research activities to support their results.

This research work based on the secondary information collected by the authors. Nevertheless some further questions can be arisen to continue this train of thought, such as:

• What are the strengths and weaknesses of the Central Highland provinces (e.g.: agriculture, tourism)?

• Are there other relevant conclusions should be drawn?

• What kind of suggestions can be formulated based on these results?

• Why did not use the authors other statistical methods as well?

The references, citations are correct. The bibliography adequately applied. Acquaintance with relevant literature and theories; review, assessment and analysis of national and international literature is very short, and not too detailed. Scientific assessment of the paper not included innovative scientific elements.

What does the Figure 1 mean? This is not a high quality (scientific) figure.

Statistics and other analyses should be performed to a higher theoretical standard and should be described in a sufficient way.

The research results are not demonstrated precisely. In some cases, there is a discrepancy between the statistics and the conclusions. The "Conclusions" section provides a brief summary of the results and discussion, but it should be more than a summary.

To sum up, without keeping up with the literature, we cannot know what other researchers are doing or contextualize our work. Staying up to date with the literature is perhaps the single most important task. In my opinion, the research results and the methodology are not high quality; therefore, this paper cannot be published in PLOS ONE.

6. PLOS authors have the option to publish the peer review history of their article (what does this mean?). If published, this will include your full peer review and any attached files.

Reviewer #1: No

Reviewer #2: No

---

## [Author Response · Author response to Decision Letter 0]

15 Jul 2021

RESPONSES TO THE REVIEWERS

We greatly appreciate the editor for giving us a chance to revise the manuscript. We are also grateful to the reviewers for their detailed valuable comments. The suggestions by the reviewers have helped us to improve the paper substantially. In the following, we present our responses to each referee’s comment. 

OUR RESPONSES TO REVIEWER #1

We wish to thank Reviewer #1 for providing many invaluable comments. We agree with most of the comments and suggestions. In the following, we present our responses to each comment.

Reviewer comment: The topic of is very interesting and also relevant. The elaboration and the applied methodology seem accurate. The tables, however, possibly should not lean to the following pages – it is just a matter of technical edition.

Authors’ response: We thank you very much for the overall assessment. Regarding the technical error of tables crossing over two pages, we have fixed in the revision version (please see Page 5,16, 18, 20, 22, 23, 24)

Reviewer comment: The bibliographic review is a bit narrow in depth and in broadness: In one hand, it should be broadened towards the experiences and lessons of other sub-asian countries, and in the other hand, the authors should utilize more international sources. In fact, in the paper in its present form there is only one non-Vietnamese source (Porter, 2008), which is far insufficient in a high-ranking international journal. The international bibliography is really rich in this respect (e.g., https://www.ersj.eu/journal/2104/download.

Authors’ response: We thank you very much for the insightful comments that help us review broader literature from international sources. We have considered your comments carefully and revised the “theoretical background” section accordingly. In particular, we reviewed the relevant literature again in order to properly situate the studies on “local competitiveness” in the contemporary competitiveness literature. We have incorporated the theories and arguments of more previous studies regarding “competitiveness” from global scale to national level and local level. For example, we highlighted the work of Tomasz and Aldona (2014); Waheeduzzaman and Ryans (2015); Shekhar and Rougata (1997); Flanagan, R., et al. (2007); Peng, Lee, and Tan (2001), Srivastava, Shah, and Talha (2006); Dhingra, T., T. Singh, and A. Sinha (2009) on theory of competitiveness (please see Page 3-4 Line 68-90). Xavier Sala-i-Martin and Elsa V. Artadi (2006); World Economic Forum (2008, 2018, 2019); Dudas, T. and A. Cibula (2018) work on Global Competitiveness Index (GCI) and national competitiveness (please see Page 4-5 Line 91-107). Moreover, as reviewer’s suggestion, the experiences and lessons of ASEAN countries have been added in the revision version (please see Page 6 Line 116-125).

We hope that the revision above would add some meaningful contributions to the understanding of Competitiveness Index from global to local context in the global South. 

Reviewer comment: In the conclusion part the suggestions/proposals applicable for the further improvement are merely based on the research results and look impressive, however they could be elaborated more in details as they look a bit too general in their present form.

Authors’ response: We thank the Reviewer for the suggestions. In the revised version, we added more detail conclusion (please see Page 25 Line 477-484).

Reviewer comment: After the mentioned improvements were done, the paper can be recommended to be accepted and published.

Authors’ response: We thank you so much for the generous assessment.

OUR RESPONSES TO REVIEWER #2

We highly appreciate Reviewer #2 for pointing out some weaknesses of the paper. The comments from the Reviewer enabled us to comprehensively revise the manuscript. We considered your suggestions carefully and accommodated them wherever applicable. Our responses to your comments are shown below. 

Reviewer comment: The title of the paper is corresponded, and the abstract is relevant to the content of paper. The paper divided into several chapters. The conceptual framework and terminology that is required to present the analysis are consistent and in line with the conventions of the given scientific field.

Authors’ response: We thank you very much for the overall assessment

Reviewer comment: A research hypothesis is a specific, clear, and testable proposition or predictive statement about the possible outcome of a scientific research study. A hypothesis is a tentative statement about the relationship between two or more variables. Specifying the research hypotheses is one of the most important steps in a scientific research study. This publication does not contain any hypotheses.

Authors’ response: We thank you very much for the insightful comments that help us to develop clear hypotheses for the paper. We acknowledge that the original version lacked hypotheses. We have considered your comments carefully and revised the manuscript. Especially, in the end of theoretical background section, we have added the information as suggested. Particularly, it is written as follow: “The hypothesis of this research is that PCI scores will be improved when ten indicators become better, additionally, the higher province’s PCI scores are, the more enterprises will be attracted and developed in the province (Figure 1)”

Figure 1. The hypotheses of research 

Reviewer comment: This research work is a secondary data analysis, as was written in the “Methodology" section. The lone primary research result of the paper is the correlation analysis. The authors should use more intensive primary research activities to support their results.

Authors’ response: We are very grateful for the valuable comments. We acknowledge this weakness in the original version as we did not explain details about the source of data nor data gathering method, in particular regarding the primary data we collected from the managers of enterprises and provincial governments. In this regard, this study both qualitative and quantitative methods were applied to investigate the relationship between PCI and enterprise attraction in the Central Highland, Vietnam. In section 3.1 of the revised version, we added more details on the sources of data we collected and the data analysis methods (Please see Page 9-10 Line 186-220).

During the fieldwork we carried out 2 focus group discussions (with the attendant of 5-8 participants for each one) and 8 in-depth interviews with local authorities in two case study provinces namely Dak Nong and Lam Dong. Such information is useful for us to figure out the attitude and evaluation of firm owners with business environment and the quality of economic governance in the province; and the views and opinions of the provincial governments in term of administrative reform, policy supports, and the transparent business environment in the province.

The result findings from fieldwork also have been added in the Results and Discussion section (Please see Page 13 Line 281-283) and (Please see Box 1 Page 21). 

Reviewer comment: What are the strengths and weaknesses of the Central Highland provinces (e.g.: agriculture, tourism)?

Authors’ response: We thank the Reviewer for the suggestions. We realize this limitation in the original version as we did not explain details about the strengths and weaknesses of the Central Highland provinces as well as the reason why Central Highland is selected for study site. In the revised version, we added descriptions about such information. This argument has been rewritten in the revised version (Please see Page 2 Line 36-43).

Reviewer comment: Are there other relevant conclusions should be drawn?

Authors’ response: We thank you for the comment. In the revised version, we have extended our conclusion part by synthesizing the main findings of the research and proposing several policy recommendations (Please see Page 25 Line 477-484)

Reviewer comment: What kind of suggestions can be formulated based on these results?

Authors’ response: We thank you for this comment. However, the suggestions were mentioned in the Conclusion and Recommendation section of manuscript as follows: “In order to improve competitiveness and further to promote enterprise attraction, it is necessary to implement solutions in coming years in the region including: (i) Continue to further improve and more strongly reform public administrative procedures, with a focus on simplification of administrative procedures, increasing online public services, through which minimizes the waiting time and informal charges of businesses; (ii) Reform vocational training system (vocational school system, vocational training programs and methods) towards better meeting the needs of the labor market, especially the needs of enterprises in the region; (iii) Renovate and enhance role of judicial system: on the one hand, promoting the role of commercial arbitration centers in resolving corporate disputes to avoid overloading of courts and shorten time, on the other hand to increase the court's support for commercial arbitration centers to ensure state power in dispute solving”.

Reviewer comment: Why did not use the authors other statistical methods as well?

Authors’ response: We thank you for this comment. You and the other reviewer commented on such content. However, your impressions on statistical methods appear to differ from other reviewer who think that the study methods are appropriate to illustrate the foci of the debate. We respond your question as follow. In the 4.3 section, we greatly believe that correlation analysis is the best choice in our case to clarify the relationship between PCI points, its sub-indexes and the attraction of enterprises in each province and the whole region. 

In fact, we initially intended to use regression models, in which dependent variable are one of three indicators: the number of enterprises, the number of labor of enterprises, and the capital of enterprises. On the other hand, the independent variables are PCI and its sub-indices scores in PCI report database in the 2008-2019 period. However, the adoption of regression models in such case would face with two issues. First, there should not exist actual causal relationship between the number of enterprises, the number of labor of enterprises the capital of enterprises and PCI and its sub-indices scores, thus It is almost impossible to predict a direct cause and effect in regression analysis. The second issue is relevant to the limitation of database source as we are only able to extract the PCI and its sub-indices on the last ten years. Thus, when we run the regression models, the results would not show the significant statistics and could not explain precisely the reality. 

Moreover, in existing version of the manuscript, the outputs from correlation analysis help us to clarify clearly the relationship between PCI and the enterprise development in the study site. Thus, we decided to retain such statistical methods in the revision version. 

Reviewer comment: The references, citations are correct. The bibliography adequately applied. Acquaintance with relevant literature and theories; review, assessment and analysis of national and international literature is very short, and not too detailed. Scientific assessment of the paper not included innovative scientific elements.

Authors’ response: We thank you very much for the insightful comments that help us review broader literature from international sources. We have considered your comments carefully and revised the “theoretical background” section accordingly. In particular, we reviewed the relevant literature again in order to properly situate the studies on “local competitiveness” in the contemporary competitiveness literature. We have incorporated the theories and arguments of more previous studies regarding “competitiveness” from global scale to national level and local level. For example, we highlighted the work of Tomasz and Aldona (2014); Waheeduzzaman and Ryans (2015); Shekhar and Rougata (1997); Flanagan, R., et al. (2007); Peng, Lee, and Tan (2001), Srivastava, Shah, and Talha (2006); Dhingra, T., T. Singh, and A. Sinha (2009) on theory of competitiveness (please see Page 3-4 Line 68-90). Xavier Sala-i-Martin and Elsa V. Artadi (2006); World Economic Forum (2008, 2018, 2019); Dudas, T. and A. Cibula (2018) work on Global Competitiveness Index (GCI) and national competitiveness (please see Page 4-5 Line 91-107). Moreover, as reviewer’s suggestion, the experiences and lessons of ASEAN countries have been added in the revision version (please see Page 6 Line 116-125).

We hope that the revision above would add some meaningful contributions to the understanding of Competitiveness Index from global to local context in the global South.

Reviewer comment: What does the Figure 1 mean? This is not a high quality (scientific) figure.

Authors’ response: We realize that the Figure 1 was not a good choice. We have revised it for easier understanding in the revision version (please see Page 8 Figure 1). 

Reviewer comment: Statistics and other analyses should be performed to a higher theoretical standard and should be described in a sufficient way.

Authors’ response: We greatly appreciate this comment from the Reviewer. The comment is repeated in the comment above. Thus, we already responded to it respectively above.

Reviewer comment: The research results are not demonstrated precisely. In some cases, there is a discrepancy between the statistics and the conclusions. 

Authors’ response: Thank you so much for catching these glaring and confusing errors, which we have corrected. We have gone through the entire manuscript carefully and checked every sentence to correct technical errors, avoid obscure meaning, and clarify our meaning. 

Reviewer comment: The "Conclusions" section provides a brief summary of the results and discussion, but it should be more than a summary.

Authors’ response: We thank the Reviewer for the suggestions. In the revised version, we added more detail conclusion to look over the broader context of the PCI and proposed some policy recommendation for local government to improve competitive capacity and attract more enterprises investing in the region (please see Page 25 Line 477-485).

Reviewer comment: To sum up, without keeping up with the literature, we cannot know what other researchers are doing or contextualize our work. Staying up to date with the literature is perhaps the single most important task. In my opinion, the research results and the methodology are not high quality; therefore, this paper cannot be published in PLOS ONE.

Authors’ response: We thank the Reviewer for the suggestions. We have tried our best to improve the manuscript based on the comments from both Reviewers. We believe that the manuscript has been improved substantially. Hence, we are always welcome to receive further constructive comments from Reviewer and the Editor that help us amend the paper to meet high standard of the journal.

---

## [Decision Letter · Decision Letter 1]

9 Aug 2021

Effect of Provincial Competitiveness Index on Enterprise Attraction in the Central Highlands, Vietnam

PONE-D-21-15678R1

Dear Dr. Nguyen Phuong Le,

We’re pleased to inform you that your manuscript has been judged scientifically suitable for publication and will be formally accepted for publication once it meets all outstanding technical requirements.

Kind regards,

László Vasa, PhD

Academic Editor

PLOS ONE

Additional Editor Comments (optional):

Reviewers' comments:

Reviewer's Responses to Questions

**Comments to the Author**

1. If the authors have adequately addressed your comments raised in a previous round of review and you feel that this manuscript is now acceptable for publication, you may indicate that here to bypass the “Comments to the Author” section, enter your conflict of interest statement in the “Confidential to Editor” section, and submit your "Accept" recommendation.

Reviewer #1: All comments have been addressed

Reviewer #2: All comments have been addressed

2. Is the manuscript technically sound, and do the data support the conclusions?

Reviewer #1: Yes

Reviewer #2: Yes

3. Has the statistical analysis been performed appropriately and rigorously? 

Reviewer #1: Yes

Reviewer #2: Yes

4. Have the authors made all data underlying the findings in their manuscript fully available?

Reviewer #1: Yes

Reviewer #2: Yes

5. Is the manuscript presented in an intelligible fashion and written in standard English?

Reviewer #1: Yes

Reviewer #2: Yes

6. Review Comments to the Author

Reviewer #1: (No Response)

Reviewer #2: The earlier mentioned improvements have been done.

The review, assessment and analysis of national and international literature have been significantly improved.

The quality of research results and the methodology are substantially higher after the revision.

In this form the paper can be recommended to be accepted and published.

7. PLOS authors have the option to publish the peer review history of their article (what does this mean?). If published, this will include your full peer review and any attached files.

Reviewer #1: No

Reviewer #2: No

---

## [Editor Report · Acceptance letter]

1 Sep 2021

PONE-D-21-15678R1 

Effect of Provincial Competitiveness Index on Enterprise Attraction in the Central Highlands, Vietnam 

Dear Dr. Le:

I'm pleased to inform you that your manuscript has been deemed suitable for publication in PLOS ONE. Congratulations! Your manuscript is now with our production department. 

Kind regards, 

on behalf of

Prof. Dr. László Vasa 

Academic Editor

PLOS ONE